# Heart Rate Variability, Blood Pressure and Peripheral Oxygen Saturation during Yoga *Adham* and *Mahat* Breathing Techniques without Retention in Adult Practitioners

**DOI:** 10.3390/jfmk9040184

**Published:** 2024-10-03

**Authors:** David Catela, Júlia Santos, Joana Oliveira, Susana Franco, Cristiana Mercê

**Affiliations:** 1Sport Sciences School of Rio Maior, Santarém Polytechnic University, Avenue Dr. Mário Soares No. 110, 2040-413 Rio Maior, Portugal; sfranco@esdrm.ipsantarem.pt (S.F.); cristianamerce@esdrm.ipsantarem.pt (C.M.); 2Quality Education-Life Quality Research Centre (CIEQV), Santarém Polytechnique University, Complex Andaluz, Apart 279, 2001-904 Santarém, Portugal; joanafpy1@gmail.com; 3Sport Physical Activity and Health Research & Innovation Center (SPRINT), Santarém Polytechnic University, Complex Andaluz, Apart 279, 2001-904 Santarém, Portugal; 4Santarém Higher School of Health, Santarém Polytechnic University, Quinta do Mergulhão Senhora da Guia, 2005-075 Santarém, Portugal; julia.santos@essaude.ipsantarem.pt; 5Individual and Community Health-Life Quality Research Centre (CIEQV), Polytechnique University of Santarém, Complex Andaluz, Apart 279, 2001-904 Santarém, Portugal; 6Portuguese Yoga Federation (FPY), Campo Emílio Infante da Câmara, 2000-014 Santarém, Portugal; 7Physical Activity and Health-Life Quality Research Centre (CIEQV), Polytechnique University of Santarém, Complex Andaluz, Apart 279, 2001-904 Santarém, Portugal; 8Interdisciplinary Center for the Study of Human Performance (CIPER), Faculty of Human Kinetics, University of Lisbon, Cruz Quebrada-Dafundo, 1499-002 Lisboa, Portugal

**Keywords:** breathing control, yoga, pranayama, heart rate variability, psychological states, motor control, fitness, non-pharmacological therapy

## Abstract

**Background:** Heart rate variability (HRV) is the change in time intervals between heart beats, reflecting the autonomic nervous system’s ability to adapt to psychological and physiological demands. Slow breathing enhances parasympathetic activity, increasing HRV. *Pranayama*, a yoga breathing technique, affords the conscious regulation of respiration frequency. This study aimed to characterize HRV, blood pressure and peripheral oxygen saturation of basic yoga breathing slow techniques with regular yoga practitioners. **Methods:** In total, 45 yoga practitioners were included in the study (including 7 males, mean age of 54.04 ± 11.97 years) with varying levels of yoga experience (minimum 3 months, maximum 37 years). Participants performed three breathing conditions: baseline (control) and two yoga techniques (abdominal (*adham*) and complete (*mahat*)) breathing, each for 10 min in the supine position (i.e., *savasana*). For each condition, respiratory frequency, heart rate (HR), blood pressure and peripheral oxygen levels were collected. **Results:** The findings revealed that both abdominal and complete yoga breathing techniques promoted a decrease in respiratory frequency (*p* < 0.001, r = 0.61; *p* < 0.001, r = 0.61, respectively), and an increase in peripheral oxygen saturation (*p* < 0.001, r = 0.50; *p* < 0.001, r = 0.46, respectively), along with blood pressure decreases in all mean values, and a significant decrease in systolic pressure, considering all conditions (*p* = 0.034, W = 0.08). There were significant increases in standard deviation of HR during abdominal and complete yoga breathing techniques compared with the baseline (*p* = 0.003, r = 0.31; *p* < 0.001, r = 0.47, respectively), indicating enhanced parasympathetic activity. Moreover, the complete breathing technique exhibited the greatest variability in HRV measures, with several significant differences compared with abdominal breathing (standard deviation of HR, *p* < 0.001, r = 0.42; SD2, standard deviation of points perpendicular to the Poincaré parallel line, *p* < 0.003, r = 0.31; SD1/SD2, *p* < 0.003, r = 0.31), suggesting a more profound impact on autonomic modulation. **Conclusions:** simple, inexpensive and non-intrusive abdominal and complete yoga breathing techniques can effectively and momentarily enhance HRV and oxygen saturation in adults, mature adults and the elderly.

## 1. Introduction

Heart rate (HR) should have the flexibility and the quickness to adapt to sympathetic and parasympathetic modulating activity branches of the autonomous nervous system; the first being related to energy mobilization, and the latter to vegetative and restorative activity. The prolonged dominance of the sympathetic system, with an augmentation in HR frequency and the consequent reduction in heart rate variability (HRV), i.e., the variation of the time interval between heartbeats, is associated with a diversity of health problems, e.g., [1,2]. A healthy high HRV indicates the capacity of autonomic mechanisms to adapt, whereas a low HRV is indicative of an abnormal and/or insufficient capacity for autonomic system adaptation, which is likely to be associated with some form of physiological dysfunction [3]. Heart rate variability (HRV) has been quantified as an index of vagal activity at designated time points and frequencies. In the time domain, the most frequently employed measures are the standard deviation of the intervals between beats, the standard deviation of R-R intervals (SDNN) (R is a peak represented in the electrocardiogram) and the root mean square of successive differences (RMSSD). In the frequency domain, the spectral powers associated with low frequency (LF) and high frequency (HF) are of primary interest [4]. In LF, an oscillation between 0.04 and 0.15 Hz is predominantly related to vasomotor activity, and, in HF, an oscillation between 0.15 and 0.4 Hz is associated with respiratory activity. HF is regarded as an indicator of the predominancy of vagal efferent activity, whereas LF is considered to reflect sympathetic modulation. Nevertheless, it is postulated that LF and HF are the results of a continuous interaction between the sympathetic and parasympathetic branches of the autonomic nervous system (ANS) [5]; it seems plausible that HF and LF components may reflect parasympathetic activity [6]. To allow a comparison across studies, it is recommended to report the LF and the HF in both absolute and normalized forms, i.e., the natural logarithm of LF (LFn) and HF (HFn) absolute values [7]. Given the intricate interplay between the two branches of the autonomic nervous system (ANS), non-linear methodologies were devised to examine the fluctuations in heart rate (HR) in response to external stimuli. Among these techniques, the Poincaré plot has emerged as a prominent analytical tool, e.g., [8]. The scatterplot depicts a present NN interval plotted against its preceding interval, revealing an ellipsoid configuration of points around a diagonal line (the value against itself). The points situated on the aforementioned line represent the instances of HR deceleration, whereas those located below the line indicate the occurrences of HR acceleration [9]. So, the Poincaré plot can not only be analyzed qualitatively but also quantitatively, where, for the shortest radius of the ellipsoid, the standard deviation of points perpendicular to the diagonal line is estimated (SD1), which has been regarded as the equivalent of RMSSD [10]. The short-term (vagal) NN variability is represented by the longest radius of the ellipsoid, which is parallel to the designated line of identity. The standard deviation of the points is estimated (SD2), which is considered equivalent to SDNN [8], and represents the long-term (sympathetic) NN variability. A third index, SD1/SD2, with similar results to the LF/HF ratio, reflects the sympathovagal balance, e.g., [11]. Qualitatively, Poincaré plot shapes are characterized by a “comet tail” in a healthy heart, whereas, in health problems, narrow, “torpedo” and complex clustered shapes occur, e.g., [12,13].

Slow-paced breathing has significant immediate (acute) effects on reducing systolic and diastolic blood pressure, albeit modestly [14], with an increasing heart rate variability in the time domain, namely the mean square of successive differences between normal heartbeats (RMSSD) and the standard deviation of NN intervals (SDNN), with a decreasing heart rate [15]. Slow breathing at around six cycles per minute provides synchronisation (resonance) between the respiratory and cardiovascular systems [16], with likely enhancement of the parasympathetic nervous system [17] and maximizing heart rate variability [18] and baroreflex sensitivity [19]. Breathing at a rate of six breaths per minute (0.1 Hz) means that it is in the LF band (0.04–0.15 Hz) of the spectral analysis and that changes should be more apparent, e.g., [20]. At 12 breaths per minute (0.2 Hz), it should be expected to be in the HF band (0.15–0.4 Hz), e.g., [8]. A reduction in the number of breaths per minute from 20 (0.33 Hz) to 15 (0.25 Hz) in the presence of a beta-adrenergic blockade has been observed to result in a decrease in the amplitude of low frequency (LF) and an increase in high frequency (HF) amplitude [21]. Slow breathing techniques have been demonstrated to enhance parasympathetic activity, with increased heart rate variability (HRV), thereby providing emotional control and psychological wellbeing [22]. A cumulative effect on HRV can be achieved within a 30-minute period, comprising 10 6-breaths-per-minute sessions [23]. A respiratory sinus arrhythmia of 6 breaths per minute is perceived as indicating lower levels of arousal than a rate of 12 breaths per minute [24]. Side effects (e.g., feeling anxious, having intrusive thoughts, feeling out of control) may also decrease. At a rate of 16 breaths per minute, systolic blood pressure changes, always preceded by RR interval changes, suggesting a baroreflex link independent of the sympathetic drive [25]. Even in patients with chronic heart failure, slow breathing has been shown to be beneficial for the cardiovascular system [26]. Below 10 breaths per minute, blood oxygen saturation increases, with increased baroreceptor sensitivity and significant reduction of systolic blood pressure, even at home [27], probably because slow and deep breathing allows a reduction in the high resting sympathetic tone characteristic of patients with chronic heart failure [28]. Slow deep breathing also has a positive effect on systolic and diastolic blood pressure [29,30,31]. The regular practice of deep breathing may result in long-term reductions in blood pressure, even when compared with the effects of physical activity [32]. The majority of studies examining the effects of yoga techniques on heart rate variability employ a variety of techniques [33] in accordance with the tenets of yoga philosophy. This approach, however, hinders the ability to discern the individual contributions of each technique, thereby limiting the acquisition of a comprehensive understanding of the specific effects on vital signs. Nevertheless, slow breathing yoga techniques are receiving particular attention, namely their effects on blood pressure. These studies frequently involve advanced yoga procedures, such as nostril or breath hold [34]; although, some studies have tried a more controlled approach with external pacing, namely music, e.g., [35].

Breathing as a meditative practice in yoga and an advanced practice called pranayama, through breath control and expansion with retention, allows the conscious regulation of breathing rate, depth and/or the inhalation/exhalation ratio [36]. *Prana* means breath, respiration, life, vitality, energy or strength, and *ayama* means stretch, extension, expansion, length, breadth, regulation, prolongation, restraint or control [37]. In the seminal book of the yogic philosophy, *Yoga Sutra* (II BC– III), Patanjali describes *pranayama* as the interruption of inhalation and exhalation, cf., [38], and it is classified as the fourth stage on a scale of eight branches, in which the first three are related to the domain of ethical behavior of the individual in society (*yama*), personal rules of conduct (*niyama*) and physical postures (*asana*). The same is said in the Hatha Yoga texts (*Hatha Yoga Pradipika, Gheranda Samhita* and *Shiva Samhita*), dating back to the XIV to XVIII centuries, where *pranayama* is identified with the notion of *kumbhaka* (retention). *Hatha Yoga Pradipika* teaches nine different *pranayama*, *Gheranda Samhita* I eight and *Shiva Samhita* one. In all of these books, *pranayama* refers to techniques with some degree of complexity, to be practiced under the supervision of an experienced teacher, requiring rigorous and sustained preparation in advance [39]. *Pranayama* is not regarded as a fundamental tenet of yogic philosophy. It is merely a breathing technique designed to facilitate the introduction of extra oxygen into the lungs. In contrast, *pranayama* employs breathing as a means of influencing the flow of *prana* (life force) in the energy channels, the *nadis* of the energy body, *pranamaya kosha* [40]. A complete respiratory cycle comprises four distinct phases: inhalation (*puraka*), lung inflation (*antara kumbhaka*), exhalation (*rechaka*) and lung deflation (*bahya kumbhaka*). Considering the body parts involved, the various breathing techniques can be classified into the following four distinct categories [37]: (i) high or clavicular breathing, which involves activating the upper parts of the lungs through the use of the neck the throat and sternum muscles with the upper ribs and the collar bone pulled upwards [40]; (ii) the mid intercostal thoracic region, where only the central part of the lungs is engaged by the expansion and contraction of the ribcage; (iii) *adham pranayama*, in the low abdominal/diaphragmatic region, where inhalation involves the diaphragm moving downwards and outwards, while exhalation entails the diaphragm moving upwards and abdominal contents moving inwards [40], with the lower portions of the lungs being primarily activated, whereas the upper and central portions remain less active; (iv) *mahat yoga pranayama* or *dirga pranayama*, is a complete practice that combines the three types aforementioned, using the entirety of the lungs in their fullest capacity.

Although the publishing about the effects of yoga practices on HRV and blood pressure [41] is diverse, an analysis of the acute effect of the control breathing element in accessible yoga calming techniques on HRV, blood pressure and peripheral oxygen saturation has not yet been carried out. Consequently, the purpose of this study was to characterize the acute effect of accessible yogic calming breathing control techniques on vital signs (heart rate, breathing frequency, blood pressure, peripheral oxygen saturation) and on heart rate variability. Considering the classification presented above, the low abdominal/diaphragmatic and complete techniques were specifically chosen for their simplicity, both without retentions [37,42], as they are easily accessible to regular yoga practitioners, even to beginners. Considering the previous literature of slow-paced breathing, we hypothesize that abdominal and complete yoga breathing techniques can promote a decrease in respiratory frequency and blood pressure [41,43], along with an increase in peripheral oxygen saturation and HRV [14,15,17].

## 2. Materials and Methods

### 2.1. Sample

The sample size was calculated by G∗Power software (version 3.1). Statistical probability value (α) was set at 0.05, statistical power at 0.95 and the non-sphericity correction = 0.5 [44]; the effect size was set at 0.25, as it represents the cutoff for a medium effect [45]; and the correlation between measures was set at 0.7, based on a previous study that evaluated HRV in the same three breathing conditions, i.e., normal breathing at baseline (control condition without any indication), abdominal (*adham*) and complete (*mahat*) yoga breathing. The sample size was calculated to be 44. From an initial sample of 50 volunteers, a withdrawal of 5 participants occurred, 2 of them due to individual time constraints, 2 due to the breathing conditions and 1 due to unwellness, resulting in a final sample size of 45 participants.

In order to recruit the sample, a protocol was made with the Portuguese Yoga Federation (FPY) for scientific partnership purposes, ensuring access to yoga certified instructors and regular practitioners. The recruitment of yoga practitioners was carried out by the instructors in order to ensure that practitioners acquired the two yogic breathing techniques. The convenience sample was composed of yoga practitioners (9.44 ± 9.08 years of practice, min 3 months, max 37 years), over 18 years old (54.04 ± 11.97 years old, min 32 years, max 78 years), both sexes (7 males), without exclusion criteria. Eight participants were on medication for hypertension (17.8%), seven were regular smokers (15.6%), seven had psychiatric support with medication (15.6%), five were on medication for asthma (11.1%), four were receiving hormone treatments (8.9%), three were being monitored and medicated for heart problems (6.7%), three were on medication for cholesterol (6.7%), one was on medication for diabetes (2.2%), one had diagnosed sleep apnea (2.2%), one had varicose veins (2.2%) and another was on medication for joint arthritis (2.2%). As some of the participants presented with multiple health issues and related medication regimes, a complete characterization is presented below (Table 1).

The purpose of such a diversified sample was intentional in order to test the universality of the effects of these yoga techniques with an ample spectrum of ages and health conditions. The percentage distribution among sexes was similar to that found in the Portuguese yoga practitioner census.

This project was approved by the research unit ethics committee of the Polytechnique University of Santarém (approval number 2-2023ESDRM), Portugal.

### 2.2. Experimental Design

This study employed a quasi-experimental design, with subjects serving as their own control, and employed a comparative and associative approach, with a single level of blindness.

### 2.3. Procedures and Data Treatment

Each volunteer gave informed written consent before this study, in compliance with the Helsinki treaty, and subsequent updates, for human studies. Information about years of yoga practice, health problems and medication were collected. Records were conducted at a Yoga Portuguese Federation center, with subjects in the supine position (i.e., *savasana*), in a quiet environment, with a room temperature of 19.92° (±1.45) and humidity of 56.51% (±8.85). Participants were instructed not to smoke, drink alcohol or coffee 4 h before data collection [4]. The experimental session was structured according to the following sequence: 10 min of rest with normal breathing (control condition without any indication), taken as baseline; and 10 min of each breathing technique: abdominal and complete breathing. The baseline line was always collected first, breathing techniques were alternated among participants (first abdominal = 23; first complete = 22). For the abdominal technique, participants were asked to exhale through their nostrils slowly, while gently contracting their abdomen, and then, to inhale through their nostrils while slowly expanding their abdomen. All the attention remained in the abdominal area. The complete technique was performed by stimulating the release of air through the nostrils in a slow, smooth and complete manner, maintaining the focus of attention from the clavicular region to the abdominal region (exhalation) and then, in the opposite direction, raising awareness from the abdominal region to the clavicular muscle while inhaling slowly, smoothly and completely (inhalation). The RR interval was collected using the Polar©V800 (Polar Electro Oy, Kempele, Finland), which was validated for adults with a combined error rate of 0.086% and an intraclass correlation coefficient (ICC) greater than 0.999 [46], and a chest strap Polar H10, also validated for adults [47]. Data were exported as .txt files and ectopic beats were manually excluded, e.g., [48]. HRV analysis was conducted using gHRV© software, version 1.6 [49]. Blood pressure was recorded at one-minute intervals using a Pic© Solution-Classic Check tensiometer (Grandate, Italy), which fulfils the international protocol requirements, including self-measurement of blood pressure [50]. In accordance with the International Protocol of the European Society of Hypertension [51], the following criteria must be met: age, appropriate cuff size and measurements taken on the left arm at the level of the heart. The frequency of respiratory cycles was observed directly on a minute-by-minute basis, and subsequently validated through the analysis of the respiratory sinus arrhythmia time series for the respiratory techniques. The standard method for respiratory rate measurement is to count the number of movements over a full minute period, observing both the abdominal and chest regions [52]. The counting of respiratory rate for one minute is also a reliable method for infants, e.g., [53]. In the course of our study, the respiratory rate was meticulously recorded at one-minute intervals for a minimum of ten minutes in each experimental condition. Peripheral oxygen saturation was collected from the index finger of the right hand, using a finger gauge Gima©, Oxi-10 model (Gima SpA, Gessate, Italy; ISO 9001/13485 certificates) [54,55], with a range of 70–100%, an accuracy of ±3% and a perfusion index of 0.2–20%. Gima© models are commonly used as professional wireless fingertip oximeters, and as reference instruments in studies. The peripheral oxygen saturation was recorded for a minimum of 10 min in each of the three conditions, with the data transmitted to a personal computer via wireless Bluetooth technology. Heart rate data were automatically filtered using adaptive thresholds to reject incorrect beats [49]. This method discards beats that exceed the cumulative mean threshold and eliminates data points that fall outside acceptable physiological values. Frequency domain analysis was obtained using a linear interpolation method. As a result, the filtered non-equispaced heart rate signal was obtained [56]. The signal was interpolated at a frequency of 4 Hz for the purposes of spectral analysis. The window size and time shift were 120 and 60 s, respectively. For the calculation of non-linear indexes, approximated entropy, SD1 and SD2 were obtained. The normalized power of LF and HF were estimated (e.g., LFn = LF/(VLF + LF + HF)) [23].

### 2.4. Statistical Treatment

Data were statistically treated with the program IBM-SPSS, version 27. A Shapiro–Wilk test was used to verify data normal distribution. For descriptive statistics, minimum value (min), maximum value (max), mean (mean), error of the mean (error) and standard deviation are presented. For comparisons, the Friedman test was used, with the Monte Carlo test, with effect size Kendall’s W estimation (0.1, small; 0.3, medium; greater than 0.5, strong); followed by the Wilcoxon test, with the Monte Carlo test and Bonferroni correction, in order to conduct a paired group comparison, with effect size *r* estimation. In the case of associations, the Spearman test (rho) was employed, with 95% confidence intervals (CI). Only those results that were significant and exhibited the same sign within CI were considered.

## 3. Results

### 3.1. Temperature and Humidity

Residual associations were found between humidity and vital signs collected. Temperature revealed inverse associations with systolic pressure, in baseline condition (rho = −0.333, *p* = 0.025, CI [−0.577; −0.035]); with breathing cycles per minute and peripheral oxygen saturation in the abdominal technique condition (rho = −0.331, *p* = 0.026, CI [−0.575; −0.032]; rho = −0.316, *p* = 0.039, CI [−0.569; −0.008], respectively).

### 3.2. Sex

No significant differences were identified between the sexes, except for systolic pressure at the baseline (Z = 2.317, *p* = 0.02, r = 1.64), which was observed to be lower in the female group (108.36 ± 13.45) than in the male group (121.07 ± 10.33). However, it is important to note that the sample size was very small, with only a few males included.

### 3.3. Age

As anticipated, both systolic and diastolic pressure increased with age in all conditions, except for diastolic pressure at baseline. Additionally, peripheral oxygen saturation decreased at baseline and in abdominal conditions. For breathing frequency, a significant inverse association was observed exclusively during the abdominal condition (Table 2, baseline condition).

### 3.4. Breathing Frequency

Breathing frequency (Table 3) was significantly different between conditions (χ_r_^2^(45,2) = 69.53, *p* < 0.001, W = 0.77). Paired comparisons revealed difference among all conditions (complete baseline T = 5.819, *p* < 0.001, r = 0.61; complete abdominal T = 3.309, *p* < 0.001, r = 0.35; abdominal baseline T = 5.757, *p* < 0.001, r = 0.61).

### 3.5. Blood Pressure

There was a notable discrepancy in systolic pressure (Table 3) across the various conditions (χ_r_^2^(45,2) = 6.782, *p* = 0.034, W = 0.08). However, when paired comparisons were conducted with the Bonferroni correction, the results did not reach statistical significance. Nevertheless, the mean systolic pressure was observed to be lower during the practice of yoga breathing techniques. A 62-year-old woman with a confirmed diagnosis of sleep apnea exhibited elevated mean systolic pressure, positioning her as a moderate outlier within the abdominal condition cohort.

No significant differences were observed in diastolic pressure (Table 3) between the various conditions (χ_r_^2^(45,2) = 0.483). A 66-year-old woman exhibited a moderate outlier (higher mean diastolic pressure) in the abdominal condition.

Indeed, the two female subjects who were identified as moderate outliers had 12 and 20 years of declared yoga practice, respectively, and were not registered as having probable hypertension. However, it was observed that there was an augmentation in both the mean systolic and diastolic pressures during the breathing yoga techniques in comparison with the baseline measurements. A detailed characterization of this phenomenon is presented in the table below (Table 4). 

### 3.6. Peripheral Oxygen Saturation

Due to technical problems, the data of peripheral oxygen saturation of two participants were lost.

A significant difference was observed in peripheral oxygen saturation between the various conditions (χ_r_^2^(43.2) = 22.68, *p* < 0.001, W = 0.26). Paired comparisons revealed no significant difference between the two yoga techniques (complete abdominal T = 0.350, *p* = 0.726). However, there were significant differences between these breathing techniques and the baseline (complete baseline T = 4.245, *p* < 0.001, r = 0.46; abdominal baseline T = 4.595, *p* < 0.001, r = 0.50). The breathing techniques demonstrated greater mean values of peripheral oxygen saturation, reinforced by higher minimum and maximum values, along with a lower standard deviation (Table 3).

### 3.7. Heart Rate Variability

No significant differences were identified between the sexes with regard to heart rate variability. There was an inverse association between age and almost all estimated temporal parameters, both in baseline and abdominal breath conditions. However, this pattern became less pronounced during complete breathing conditions (Table 5).

Standard deviation of heart rate (Table 6) was significantly different between conditions (χ_r_^2^(45,2) = 27.94, *p* < 0.001, W = 0.31). Paired comparisons revealed difference among all conditions (complete baseline T = 4.430, *p* < 0.001, r = 0.47; complete abdominal T = 3.957, *p* < 0.001, r = 0.42; abdominal baseline T = 2.988, *p* = 0.003, r = 0.31).

Complete breathing afforded the greatest variation around the mean heart frequency, followed by abdominal breathing (Table 6).

There were no significant differences between conditions in the proportion of pairs of normal-to-normal intervals that differed by more than 50 ms (pNN50). However, the maximum value and mean and standard deviation were found to be inferior during the breathing yoga techniques (Table 6). No significant differences were observed between conditions in rMSSD, with a slight reduction in mean and standard deviation values and a superior minimum value during yoga breathing techniques (Table 6).

A significant difference was observed in the heart rate variability triangular index between the various conditions (χ_r_^2^(45,2) = 18.20, *p* < 0.001, W = 0.20). The results of the paired comparisons indicated there were no statistically significant differences between the two yoga techniques (complete abdominal T = 1.930, *p* = 0.054). However, there were significant differences between these techniques and the baseline (complete baseline T = 3.950, *p* < 0.001, r = 0.42; abdominal baseline T = 3.443, *p* < 0.001, r = 0.36), with superior minimum and mean values, which were further supported by a lower standard deviation (Table 6).

The SDNN (Table 6) exhibited marked differences between the experimental conditions (χ_r_^2^(45,2) = 26.83, *p* < 0.001, W = 0.30). The results of the paired comparisons indicated there were significant differences among all conditions (complete baseline T = 4.248, *p* < 0.001, r = 0.45; complete abdominal T = 2.746, *p* = 0.006, r = 0.30; abdominal baseline T = 3.640, *p* < 0.001, r = 0.38). The complete technique was observed to afford the greatest SDNN, with the abdominal technique (Table 6) demonstrating a similar but less pronounced effect.

The results of the SD1 analysis revealed no statistically significant differences between the conditions (χ_r_^2^(45,2) = 0.235, *p* = 0.889), with a slight reduction in the standard deviation. SD2 was significantly different between conditions (χ_r_^2^(45,2) = 33.419, *p* < 0.001, W = 0.37) and SD1/SD2 ratio (χ_r_^2^(45,2 = 14.782, *p* < 0.001, W = 0.16). The results of the paired comparisons for SD2 demonstrated a statistically significant difference between the two yoga techniques (complete abdominal T = 2.929, *p* < 0.003, r = 0.31), and a statistically significant difference between these two techniques and the baseline (complete baseline T = 4.750, *p* < 0.001, r = 0.50; abdominal baseline T = 4.205, *p* < 0.001, r = 0.44). Additionally, the complete technique exhibited superior minimum, maximum and mean values, and a lower standard deviation, which was more pronounced in the complete technique. For the SD1/SD2 ratio, paired comparisons revealed a significant difference between the two yoga techniques (complete abdominal T = 2.670, *p* = 0.008, r = 0.28), and a significant difference between these and the baseline (complete baseline T = 3.781, *p* < 0.001, r = 0.40; abdominal baseline T = 2.895, *p* = 0.004, r = 0.31), with inferior maximum and mean values. This was reinforced by a lower standard deviation, which was consistent with the expected greater accentuation in the complete technique (Table 7).

A significant difference was observed in the very low frequency (VLF) between the various conditions (χ_r_^2^(45,2) = 28.458, *p* < 0.001, W = 0.32). Paired comparisons revealed no significant difference between the baseline and the abdominal technique (T = 1.800, *p* = 0.072), However, there was a significant difference between these and the complete technique (complete baseline T = 5.135, *p* < 0.001, r = 0.54; complete abdominal T = 3.968, *p* < 0.001, r = 0.42); with superior minimum, maximum and mean values for both yoga breathing techniques (Table 7).

Low frequency (LF) was significantly different between the conditions (χ_r_^2^(45,2) = 44.592, *p* < 0.001, W = 0.50). The results of the paired comparisons indicated no statistically significant difference between the two yoga techniques (complete abdominal T = 1.315, *p* = 0.189) and the baseline. Nevertheless, there were significant differences between these techniques and the baseline (complete baseline T = 4.820, *p* < 0.001, r = 0.51; abdominal baseline T = 5.006, *p* < 0.001, r = 0.53), with superior minimum and mean values, which were further supported by a lower standard deviation (Table 7).

With regard to SD1 and rMSSD, no notable discrepancies were observed in the high frequency (HF) between the experimental conditions (χ_r_^2^(45,2) = 4.570, *p* = 0.102); although, there was a notable decline in the mean and standard deviation values, with the minimum value exhibiting superiority during both yoga breathing techniques (Table 7).

The LF/HF ratio was significantly different between conditions (χ_r_^2^(45,2) = 44.726, *p* < 0.001, W = 0.50). The results of the paired comparisons revealed no statistically significant difference between the two yoga techniques (complete abdominal T = 0.480, *p* = 0.631). However, there was a statistically significant difference between these techniques and the baseline (complete baseline T = 5.217, *p* < 0.001, r = 0.55; abdominal baseline T = 5.311, *p* < 0.001, r = 0.56), with superior minimum and mean values, although with higher standard deviations (Table 7).

A significant difference was observed in LFn between the various conditions (χ_r_^2^(45,2) = 13.687, *p* = 0.001, W = 0.15), and HFn (χ_r_^2^(45,2) = 45.128, *p* < 0.001, W = 0.50). For LFn, paired comparisons revealed a statistically significant difference between the two yoga techniques (complete abdominal T = 3.697, *p* < 0.001, r = 0.39), and a significant difference between the abdominal technique and the baseline (T = 4.159, *p* < 0.001, r = 0.44). However, the aforementioned techniques of complete technique and the baseline (T = 0.805, *p* = 0.421) were not exhaustive. For HFn, paired comparisons revealed no significant difference between the two yoga techniques (T = 1.349, *p* = 0.177), and a notable difference existed between these techniques and the baseline (complete baseline T = 5.555, *p* < 0.001, r = 0.59; abdominal baseline T = 5.300, *p* < 0.001, r = 0.56), with inferior maximum and mean values reinforced by a lower standard deviation. The complete technique was expected to demonstrate a greater accentuation (Table 7).

The minimum, maximum and mean values for LFn were observed to be higher in the yoga technique than in the baseline and higher in the abdominal technique than in the full technique. The minimum, maximum, mean values and standard deviation for HFn were observed to be lower in the yoga techniques than in the baseline, and higher in the abdominal technique than in the complete technique.

## 4. Discussion

The present study aimed to characterize low abdominal/diaphragmatic (*adham pranayama*) and complete (*mahat yoga pranayama* or *dirga pranayama*) yoga breathing techniques through vital signs response and heart rate variability pattern in yoga practitioners. Considering the previous literature of slow-pace breathing, it was hypothesized that abdominal and complete breathing techniques can promote a decrease in respiratory frequency and blood pressure, along with an increase in peripheral oxygen saturation and HRV [14,15,17].

In the baseline condition, systolic pressure results were consistent with those expected for age and sex, e.g., [57], also during abdominal yogic breathing with inverse significant associations with blood pressure and oxygen peripheral saturation, even with significantly lower bcpm. During complete yogic breathing, the same pattern was observed, except for SpO_2_, where apparently age effect was diluted (Table 2), cf., [58].

In the baseline condition, the minimum value was 3.5 cycles per minute (Table 2), which was likely to have been achieved naturally by more experienced practitioners or yoga instructors. This was because, although the instruction was to breath normally, the data collection was conducted in a quiet supine position, similar to the *savasana* (corpse pose) yoga posture, as yoga practitioners tend to have a slower breathing rate during baseline breathing, e.g., [59]. In yoga practice, this posture is thought to relax the entire psycho/physiological system and is used to develop body awareness [40]. The respiratory rate during slow-paced abdominal breathing in meditation tends to be three–four breaths per minute (0.05–0.07 Hz), e.g., [60], or even once per minute, with a corresponding increase in very low frequency (VLF) (<0.05 Hz) [61].

In accordance, in both yoga techniques, breathing cycles below two cycles per minute were observed, particularly in the complete technique; probably because it involved the mobilization of both abdominal and thoracic segments, requiring more time to complete a breathing cycle. It was also interesting to note that lower standard deviations occurred during the yogic breathing techniques, meaning that this highly varied sample of practitioners became more homogeneous in their breathing pace. Synchronization (resonance) between cardiac and respiratory systems occurs at six breaths per minute (0.1 Hz) [62]. However, detection of individualized resonance frequency rates is recommended in decremental steps of 0.5 breaths per minute, from 6.5 breaths per minute to 4.5 breaths per minute [23].

Although mean systolic pressure was lower during yogic breathing techniques, which was in line with the hypothesis raised, maximum values augmented, indicating that not all practitioners experienced benefits from these techniques. After a more individual analysis, two women (62 and 66 years old, probably having hypertension, based on their baseline values and no report of medication or clinical report) revealed an augmentation of their systolic and diastolic pressures during the practice of the slow breathing yoga techniques. It was possible that the white coat effect was also present [63]. This effect is characterized by an acute increase in blood pressure specifically associated with a clinical context. Nevertheless, we recommend that it is convenient for yoga instructors to follow an individual impact of blood pressure during the practice of these techniques in order to detect whether the desired effect is not occurring for particular persons. Maybe, individualized resonance frequency rates should be explored in these particular cases [23].

Arterial and tissue oxygenation are primarily regulated through breathing [64,65]. Breathing frequency has effects on lungs’ gas exchange, with higher efficiency around six breaths per minute (0.1 Hz) [66]. In our study, as hypothesized, peripheral oxygen saturation augmented significantly and similarly during both yoga techniques, sustained in all the descriptive statistical values, namely with a reduction in the standard deviation (Table 3), meaning that the sample became more homogeneous, with values nearer those expected in a healthy adult, e.g., [67]. So, during breathing yoga techniques, the lungs’ gas exchanges were more efficient, affording greater opportunity for the body’s oxygen supply; although, we did not exclude the hypothesis of the presence of hypometabolic cases that have been found in yoga practitioners [64] due to the great number of participants with many years of yoga practice.

With age, heart rate variability progressively deteriorates independent of sex [68]. Older persons tend to show decreased HRV and reduced baroreflex sensitivity [69]. Nonetheless, temporal parameters of estimated heart rate variability revealed that the complete technique had an attenuating effect on age negative influence; a tendency that was already evident in the abdominal technique, where associations with age became lower than in the baseline condition (Table 5).

During bradypnea, decreased rMSSD and pNN50 were observed [70], as in our study during breathing yoga techniques (Table 6), which is usually associated with sympathetic activation. This pattern probably occurred because, in this study, the mean breathing frequency was very low, around four cycles per minute (0.05 Hz), for both yoga techniques (Table 3), so within the LF spectral power (0.04–0.15 Hz). Paced breathing within the lower end of the LF range [61] was under sympathetic control [71]. However, in these yoga conditions, they were almost entirely vagally mediated [72], probably with vascular tone baroreflex-specific resonance [73]. Slow breathing at six cycles per minute acutely increased baroreflex gain in healthy individuals and patients with chronic heart failure [65]. It may be that many of our practitioners were also increasing in baroreflex sensitivity through the practice of these yoga techniques. The baroreflex is a homeostatic mechanism that modulates blood pressure; when blood pressure increases it reduces the heart rate, and when blood pressure decreases it causes the heart rate to increase [74]. In this study, systolic pressure decreased during yoga breathing techniques (Table 3), a phenomenon that may reinforce the hypothesis that spontaneous baroreflex sensitivity was present during the practice of these yoga breathing techniques.

While rMSSD corresponded to short-term variability, SDNN and HRVi measured total heart rate variability [8]. Our results revealed that, contrary to rMSSD, both SDNN and HRVi were significantly higher during the yoga breathing techniques, e.g., [17], particularly during the complete technique.

The SD1 results were comparable to those of rMSSD, exhibiting a total direct association in the baseline and abdominal conditions, and a particularly robust association in the complete condition (rho = 0.939, *p* < 0.001, CI 0.889–0.967). SD1 employed an identical methodology for the calculation of rMSSD, establishing a direct correlation between the baseline and abdominal conditions, and a particularly robust association in the complete condition [9]. Similar to the study of Guzik, Piskorski, Krauze, Schneider, Wesseling, Wykretowicz and Wysocki [8], the higher breathing frequency at baseline revealed no significant changes in these two parameters of heart rate variability relative to the two yoga techniques. SD1 and rMSSD also revealed strong direct associations with LF frequency (<0.001) in all conditions (statistics are not presented), although without differences among conditions. It may be because, in the supine position, LF is particularly affected by vagal activity [4], which was the participants’ position in our study, cf., [8]. In our sample, for all the conditions, SD1 and SD2 had a strong direct association with LF and HF (<0.001; statistics not presented), suggesting balanced sympathetic and parasympathetic activity; however, there were higher heart rate variability parameters (SDNN, HRVi) during the yoga breathing techniques compared with the baseline.

There is a case study that reports that very slow breathing with a frequency of less than 0.04 Hz (2.4 breaths per minute) results in an increase in the very low frequency (VLF) band (0.003–0.04 Hz), with a significant reduction in HF power and no change in the LF/HF ratio. In our study, we found confirmation of this phenomenon. Although, at baseline there was no significant association between mean respiratory frequency and VLF, inverse significant associations were found during abdominal and complete techniques (rho = −0.519, *p* < 0.001, CI −0.710; −0.258; rho = −0.068, *p* < 0.001, CI −0.769; −0.376, respectively), meaning that other resonance modes could exist at very low respiratory frequencies.

The resonance frequency of the baroreflex system lies within the LF range; so, LF power increases when the respiration rate is in the resonance frequency and more effectively stimulates the baroreflex [73,75]. Further, cardiac vagal activity increases when individuals engage in slow-paced breathing within the LF range [72]. It appears that this LF range was predominantly where our participants were during the yoga breathing techniques, because LF, contrary to HF, was significantly higher during yoga breathing techniques than during baseline (Table 7), with LF predominance confirmed by the LF/HF ratio (Table 7) [17,76,77]. This LF predominance suggests that participants were “exercising” their vascular tone baroreflex (equally) during yoga breathing techniques [23]. As expected, SD1/SD2 and LF/HF ratios were inversely significantly associated during baseline (rho = −0.529, *p* < 0.001, CI −0.716, −0.270), complete technique (rho = −0.437, *p* = 0.003, CI −0.653, −0.156) and, particularly, abdominal technique (rho = −0.651, *p* < 0.001, CI −0.796, −0.435) [78].

The LF is primarily associated with sympathetic modulation when expressed as normalized units [4]. The results of this study indicate that, in both yoga techniques, HFn was reduced in comparison with the baseline, and LFn was augmented. These findings support the hypothesis that there was an increase in cardiac vagal activity [72], particularly during the abdominal technique.

The results of the statistical treatments consistently demonstrated the pronounced acute impact of selected yoga breathing techniques on HRV. This was thought to be due to probable reinforcement of parasympathetic activity and spontaneous baroreflex sensitivity, e.g., [8,79], as evidenced by the increase in values of parameters associated with them, such as SD2, SD1/SD2 ratio, F and LF/HF ratio. Additionally, there was a notable rise in peripheral oxygen saturation.

Paced breathing at an appropriate resonance frequency can stimulate the autonomic baroreflex function [18]. In this study, we concentrated on the spectral peak of the LF component of HRV, as the assessment of the LF component during rest should provide valuable insight into individual resonance profiles. For instance, LF HRV is associated with the baroreflex system [74].

### 4.1. Recommendations for Future Studies

Beyond the physical benefits, yoga breathing techniques have also revealed the ability to reduce stress and anxiety [80]. Although this was not the purpose of the present study, future research should incorporate subjective measures of mental wellbeing to gain a more comprehensive understanding of the advantages of these practices.

Future research should also consider a longitudinal approach in order to examine the long-term effects of yoga techniques. It is essential to gain an understanding of the potential long-term effects of this practice and its actual impact on physiological health. (e.g., heart rate variability or blood pressure). It would be particularly valuable to explore the potential benefits for individuals with hypertension and other chronic conditions. Regular assessments at defined intervals could provide insights into the maintenance of beneficial effects and the possible need for adjustments in techniques over time.

Another promising area for investigation is the individual variability in responses to yogic breathing techniques. It is crucial to understand how factors such as age, sex, fitness level, professional occupation and health status influence the effectiveness of these practices. Future research should focus on developing personalized protocols that take these individual differences into account. Customizing techniques to meet the specific needs of each individual can maximize therapeutic benefits. Studies employing subgroup analyses and individual response can provide a solid foundation for implementing more effective and targeted yogic breathing programs.

Future research should test these effects in clinical populations with lower HRV or oxygen saturation, exploring their potential as a non-pharmacological therapy.

### 4.2. Limitations

For this study, it was not possible to access equipment that would allow the moment-to-moment synchronized recording of respiratory rate, heart rate, interval between beats, blood pressure and peripheral oxygen saturation. This would make it possible to analyze with greater accuracy, for example, the resonance between cardiac and respiratory systems, the specific effect of the relative duration of the phases of the respiratory cycle or the discussion of cut-off values for a more pronounced acute beneficial effect on vital signs. Given the diversity of experience in practicing these respiratory techniques, we admit that some more experienced practitioners automatically entered meditation, which was not controlled.

## 5. Conclusions

Our results revealed that abdominal and complete yoga breathing techniques promoted a decrease in respiratory frequency and blood pressure, along with an increase in peripheral oxygen saturation. There were significant increases in HR variability (HRV) during both yoga breathing techniques compared with the baseline, indicating enhanced parasympathetic activity. These findings sustain the hypothesis that simple easily learned yoga slow breathing techniques, which are inexpensive and non-intrusive, can help adults, mature adults and the elderly to momentarily augment heart rate variability [17] and peripheral oxygen saturation. This evidence indicates an improvement in parasympathetic activity and autonomic flexibility. The complete technique revealed superior effects; however, due to its greater complexity in motor control, it is recommended to start with the abdominal technique, as this one provides similar physiological benefits in the analyzed vital signs and is easier to acquire. We recommend that it is convenient for yoga instructors to follow individual impacts of blood pressure during the practice of these techniques in order to detect whether, for particular persons, the desired effect is not occurring. Maybe, individualized resonance frequency rates should be explored in particular cases. Generally, the accessibility and effectiveness of these practices seem to make them suitable for young adults and the elderly, affording complementary non-clinical cardiac and respiratory benefits without the need for intrusive or expensive interventions.

## Figures and Tables

**Table 1 jfmk-09-00184-t001:** Participants with health-confirmed problem characterization.

Ag	Sex	Pract	Tab	Hyp	Heart	Diab	Chol	Apnea	Var	Arth	Psych	Resp	Horm
43	male	12			X	X					X		
72	male	4		X							X		
58	male	240									X		
37	male	120	X										
73	male	24		X									
76	female	360		X									
52	female	3							X				
62	female	120						X					
42	female	18									X		
63	female	240	X										
59	female	72		X						X			
59	female	60									X		X
35	female	75	X										
56	female	72									X		
57	female	60			X								X
60	female	768	X				X						
67	female	72		X								X	
68	female	96		X								X	
43	female	48	X										
55	female	24											X
43	female	72	X										
41	female	48										X	
67	female	60										X	
75	female	156		X	X		X				X	X	X
54	female	26					X						
56	female	72	X	X									
Health Problem Frequency n (%)			7 (15.6%)	8 (17.8%)	3 (6.7%)	1 (2.2%)	3 (6.7%)	1 (2.2%)	1 (2.2%)	1 (2.2%)	7 (15.6%)	5 (11.1%)	4 (8.9%)

Notes: X—presence, Ag—age, pract—yoga months of practice, tab—tabaco consumption, hyp—hypertension, heart—heart problems, diab—diabetes, chol—cholesterol, apnea—sleep apnea, var—varicose veins, arth—arthritis, psych—psychiatric problems, resp—respiratory problems, horm—hormone treatment

**Table 2 jfmk-09-00184-t002:** Association (rho, *p*; confidence intervals, CI 95%) between age and vital signs collected, per condition (baseline; abdominal technique; complete technique).

Variables	Condition	rho	*p*	Inferior CI	Superior CI
age—bcpm	Baseline	−0.197	0.195	−0.471	0.112
	Abdominal	−0.359	0.016 *	−0.596	−0.064
	Complete	−0.035	0.819	−0.333	0.269
age—syst	Baseline	0.406	0.006 *	0.119	0.631
	Abdominal	0.338	0.023 *	0.04	0.58
	Complete	0.414	0.005 *	0.128	0.636
age—diast	Baseline	0.225	0.138	−0.082	0.493
	Abdominal	0.341	0.022 *	0.044	0.583
	Complete	0.363	0.014 *	0.069	0.599
age—SpO_2_	Baseline	−0.554	<0.001 *	−0.737	−0.296
	Abdominal	−0.384	0.011 *	−0.619	−0.086
	Complete	−0.287	0.062	−0.547	0.024

Notes: bcpm—breath cycles per minute, syst—systolic pressure, diast—diastolic pressure, SpO_2_—peripheral oxygen saturation, * significant difference.

**Table 3 jfmk-09-00184-t003:** Descriptive statistics for vital signs for each condition.

Variable	Condition	Min	Max	Mean	Error	SD
bcpm *	Baseline ᵝᵓ	3.50	22.83	11.86	0.60	4.01
	Abdominal ᵝᵟ	1.50	14.75	4.88	0.35	2.37
	Complete ᵓᵟ	1.62	10.91	3.65	0.27	1.82
syst *	Baseline	84.62	140.77	110.34	2.05	13.73
	Abdominal	84.31	149.83	107.61	2.19	14.71
	Complete	85.77	144.09	107.78	2.02	13.56
diast	Baseline	46.69	85.46	66.88	1.12	7.53
	Abdominal	48.00	94.75	67.05	1.33	8.89
	Complete	50.62	89.75	66.99	1.27	8.52
SpO_2_ *	Baseline ᵝᵓ	93.52	98.78	96.81	0.19	1.26
	Abdominal ᵝ	95.19	98.98	97.67	0.14	0.91
	Complete ᵓ	95.45	98.99	97.59	0.15	1.01

Notes: bcpm—breath cycles per minute, syst—systolic pressure, diast—diastolic pressure, SpO_2_—peripheral oxygen saturation, significant difference considering all conditions *, ᵝ between baseline and abdominal conditions, ᵓ between baseline and complete conditions, ᵟ between abdominal and complete conditions.

**Table 4 jfmk-09-00184-t004:** Mean of systolic and diastolic pressures (mean breathing cycles per minute) for each condition in the two women moderate outliers.

Subjects/Condition	Baseline	Abdominal	Complete
Female. 62 years old	135.38/71.31(7.08)	149.83/84.00(2.05)	144.09/79.75(2.00)
Female. 66 years old	140.77/85.46(11.92)	144.67/94.75(3.27)	142.18/89.75(2.45)

**Table 5 jfmk-09-00184-t005:** Association (rho, *p*; confidence intervals, CI 95%) between age and estimated temporal parameters per condition (baseline; abdominal technique; complete technique).

Condition	Variables	rho	*p*	Inferior CI	Superior CI
Baseline	mean	0.067	0.662	−0.240	0.361
	sd	−0.480	<0.001 *	−0.682	−0.208
	RR interval	−0.077	0.614	−0.370	0.230
	SDNN	−0.448	0.002 *	−0.660	−0.169
	pNN50	−0.525	<0.001 *	−0.714	−0.265
	rMSSD	−0.471	0.001 *	−0.676	−0.197
	hrvi	−0.497	<0.001 *	−0.695	−0.230
Abdominal	mean	0.154	0.314	−0.155	0.435
	sd	−0.318	0.033 *	−0.566	−0.019
	RR interval	−0.153	0.316	−0.434	0.156
	SDNN	−0.375	0.011 *	−0.608	−0.083
	pNN50	−0.498	<0.001 *	−0.695	−0.231
	rMSSD	−0.442	0.002 *	−0.656	−0.162
	hrvi	−0.313	0.036 *	−0.562	−0.013
Complete	mean	0.075	0.626	−0.232	0.368
	sd	−0.222	0.143	−0.491	0.085
	RR interval	−0.089	0.560	−0.381	0.218
	SDNN	−0.301	0.045 *	−0.553	0.001
	pNN50	−0.432	0.003 *	−0.649	−0.150
	rMSSD	−0.312	0.037 *	−0.561	−0.012
	hrvi	−0.250	0.098	−0.513	0.056

Notes: mean—mean heart rate, sd—standard deviation of heart rate, RR interval—mean rr interval, SDNN—standard deviation RR interval, pNN50—proportion of NN50 divided by the total number of RR intervals, rMSSD—root mean square of successive differences, HRVi—integral of the density distribution, * significant difference.

**Table 6 jfmk-09-00184-t006:** Descriptive statistics for estimated temporal parameters, standard deviation of heart rate, proportion of NN50 divided by the total number of RR intervals, root mean square of successive differences, heart rate variability triangular index and standard deviation RR interval, SD1, SD2 and SD1/SD2, for each condition.

Variable	Condition	Min	Max	Mean	Error	SD
sd *	Baseline ᵝᵓ	1.11	7.59	3.29	0.22	1.47
	Abdominal ᵝᵟ	1.71	7.63	3.67	0.20	1.35
	Complete ᵓᵟ	1.63	8.74	4.25	0.25	1.67
pNN50	Baseline	0.00	93.00	12.27	2.82	18.91
	Abdominal	0.00	43.41	9.86	1.82	12.23
	Complete	0.00	47.71	8.48	1.68	11.29
rMSSD	Baseline	5.39	198.17	33.61	4.86	32.62
	Abdominal	7.53	156.24	32.23	3.83	25.67
	Complete	9.00	171.14	32.15	4.17	27.95
HRVi *	Baseline ᵝᵓ	3.71	26.89	11.36	0.74	4.97
	Abdominal ᵝᵟ	7.36	27.50	13.09	0.71	4.73
	Complete ᵓᵟ	6.90	26.89	14.12	0.70	4.70
SDNN *	Baseline ᵝᵓ	12.23	180.39	48.25	4.51	30.26
	Abdominal ᵝ	23.22	179.15	53.82	4.20	28.17
	Complete ᵓ	22.44	178.96	58.94	4.24	28.46
SD1	Baseline	3.82	140.23	23.78	3.44	23.08
	Abdominal	5.33	110.57	22.81	2.71	18.17
	Complete	6.54	121.11	23.13	2.93	19.62
SD2 *	Baseline ᵝᵓ	16.73	213.32	63.38	5.52	37.03
	Abdominal ᵝᵟ	31.85	228.16	72.24	5.40	36.21
	Complete ᵓᵟ	31.07	221.85	79.68	5.36	35.98
SD1/SD2 *	Baseline ᵝᵓ	0.09	0.72	0.35	0.02	0.13
	Abdominal ᵝᵟ	0.13	0.5	0.30	0.02	0.11
	Complete ᵓᵟ	0.12	0.55	0.27	0.01	0.09

Notes: Sd—standard deviation of heart rate, pNN50—proportion of NN50 divided by the total number of RR intervals, rMSSD—root mean square of successive differences, HRVi—integral of the density distribution, SDNN—standard deviation RR interval; SD1—standard deviation of points perpendicular to the Poincaré diagonal line, SD2—standard deviation of points perpendicular to the Poincaré parallel line, SD1/SD2—ratio between SD1 and SD2, significant difference * considering all conditions, ᵝ between baseline and abdominal conditions, ᵓ between baseline and complete conditions, ᵟ between abdominal and complete conditions.

**Table 7 jfmk-09-00184-t007:** Descriptive statistics for estimated frequency parameters, very low frequency, low frequency, high frequency and low/high frequency ratio, low frequency with natural algorithm and high frequency with natural algorithm for each condition.

Variable	Condition	Min	Max	Mean	Error	SD
VLF *	Baseline	3.73	586.97	121.66	180.75	125.80
	Abdominal	20.67	4874.80	319.65	122.25	820.06
	Complete	23.28	4066.98	524.80	100.94	677.10
LF *	Baseline	13.35	10,074.27	598.30	238.31	1598.63
	Abdominal	67.23	7546.51	1033.90	206.61	1385.95
	Complete	34.88	7194.92	942.31	184.97	1240.82
HF	Baseline	2.21	4460.16	295.43	107.57	721.62
	Abdominal	5.68	1816.51	158.08	42.86	287.54
	Complete	8.21	2106.15	184.19	55.77	374.09
LF/HF *	Baseline	0.26	13.35	3.6	0.44	2.98
	Abdominal	1.62	50.61	12.3	1.67	11.23
	Complete	2.42	43.38	12.3	1.41	9.43
LFn *	Baseline	0.14	0.84	0.51	0.03	0.17
	Abdominal	0.27	0.91	0.70	0.03	0.17
	Complete	0.19	0.87	0.54	0.03	0.21
HFn *	Baseline	0.06	0.68	0.26	0.02	0.17
	Abdominal	0.01	0.32	0.10	0.01	0.07
	Complete	0.01	0.25	0.08	0.01	0.06

Notes: VLF—very low frequency, LF—low frequency, HF—high frequency, LF/HF—low/high frequency ratio, LFn—low frequency with natural algorithm, HFn—high frequency with natural algorithm, significant differences considering all conditions *.

## Data Availability

Data availability is possible upon request and with the approval of institutional ethics committee.

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
