# Peer review of "Heart Rate Variability, Blood Pressure and Peripheral Oxygen Saturation during Yoga Adham and Mahat Breathing Techniques without Retention in Adult Practitioners"

_jfmk, 2024, doi:10.3390/jfmk9040184_

Round 1
Reviewer 1 Report
Comments and Suggestions for Authors
Dear Authors and Editors
Thank you for the opportunity to review this manuscript.
The manuscript focuses on evaluating the impact of selected yoga breathing techniques (Adham Pranayama, Mahat Yoga Pranayama, or Dirga Pranayama) on blood pressure, peripheral oxygen saturation, and heart rate variability patterns. This is a highly intriguing topic, filling a research gap concerning the assessment of specific breathing techniques on important parameters indicative of the adaptive capabilities of the nervous system.
The study is well-designed and executed, but there are areas where improvements can be made, particularly in structuring the abstract, introduction, discussion, and references. These adjustments will help enhance the manuscript's clarity and impact.
Abstract
I suggest the authors focus more on presenting the results, including p-values and, if possible, effect sizes or other statistical analysis values. The background section can be shortened slightly. The last sentence in the abstract can be abbreviated by removing the fragment “these results suggest.”
Introduction
The introduction serves as a good theoretical foundation for the research topic. However, the literature review could be enhanced to better highlight which aspects of this topic have already been studied, by whom, etc. The study's objective should be rephrased to remove terms like “to contribute” and “in this article.” Additionally, it would be beneficial to include a research hypothesis, especially since the authors refer to the hypothesis in the conclusion section. The sentence in lines 159-160 is out of place in this section and should be moved to the Material & Methods section.
Material & Methods
The study group was very diverse in terms of age, years of yoga practice, and comorbidities. This selection was intentional. Given this, were those who had practiced yoga for the shortest time already familiar with the breathing techniques used in this study?
Results
The results are clearly presented in seven tables and described in the text.
Discussion
The text in lines 416-419 should be removed. It would be helpful to start the discussion by reiterating the study's objective. Throughout the discussion, refer back to the objective and hypotheses (which should be added as mentioned above). Referring to tables in the discussion seems unnecessary. Alongside recommendations for future studies, it would also be useful to add a section on the limitations of the current study.
Conclusions
I propose that in the conclusions section, the authors should not refer to other authors' studies (line 567) but rather focus solely on the conclusions drawn from their own research. The sentence in lines 574-576 is unnecessary in this section and should be placed in the “Recommendations for future studies” section.
References
The bibliography includes as many as 90 references, which is excessive for research papers. It may be worthwhile to reduce the number of references, particularly those from before the year 2000.
Author Response
Dear esteemed reviewer,
We would like to express our sincere gratitude for your invaluable feedback and insightful comments, which have significantly enhanced and enriched the manuscript.
We have diligently considered all of your comments and have made the corresponding revisions. Please be advised that the following text contains our responses to each of your comments, presented individually. To aid the review process, we have distinguished our responses and the corresponding changes in the manuscript by highlighting them in blue.
Thank you for the opportunity to review this manuscript.
The manuscript focuses on evaluating the impact of selected yoga breathing techniques (Adham Pranayama, Mahat Yoga Pranayama, or Dirga Pranayama) on blood pressure, peripheral oxygen saturation, and heart rate variability patterns. This is a highly intriguing topic, filling a research gap concerning the assessment of specific breathing techniques on important parameters indicative of the adaptive capabilities of the nervous system.
The study is well-designed and executed, but there are areas where improvements can be made, particularly in structuring the abstract, introduction, discussion, and references. These adjustments will help enhance the manuscript's clarity and impact.
Response: Thank you for your valuable comment. We are delighted that you recognize the contribution of our research to the existing literature. We have incorporated all the suggested changes as outlined in the responses to the individual comments.
Abstract
I suggest the authors focus more on presenting the results, including p-values and, if possible, effect sizes or other statistical analysis values. The background section can be shortened slightly. The last sentence in the abstract can be abbreviated by removing the fragment “these results suggest.”
Response: As per the recommendations, the background section in the abstract has been condensed, p-values and effect size have been introduced in the results, and the final sentence has been shortened (lines 24-25, 31-37).
Introduction
The introduction serves as a good theoretical foundation for the research topic. However, the literature review could be enhanced to better highlight which aspects of this topic have already been studied, by whom, etc. The study's objective should be rephrased to remove terms like “to contribute” and “in this article.” Additionally, it would be beneficial to include a research hypothesis, especially since the authors refer to the hypothesis in the conclusion section. The sentence in lines 159-160 is out of place in this section and should be moved to the Material & Methods section.
Response: Thank you for your recommendations. We have revised the study objectives and have omitted the phrases "to contribute" and "in this article,” as per your suggestion. In addition, we have incorporated a research hypothesis supported by relevant previous literature as you advised (lines 171-174). We have replaced the phrase identified in the "Sample" subsection of the "Materials and Methods. The existing studies on yogic breathing techniques employ a wide range of methodologies and assess their impact on vital signs. These studies, which frequently incorporate postural and meditative techniques in addition to breathing techniques, prompted us to reference studies that could contribute to could contribute to analyzing and discussing the results obtained in our study. We have chosen not to conduct a systematic review or meta-analysis, as previous reviews have already addressed the issue of the diversity of experimental designs, frequently encompassing multiple independent variables. Our study aims to avoid this issue by adhering to the experimental design we have outlined, by our primary goal of characterizing the impact of universally accessible techniques on young adults, adults and seniors on key vital signs.
Material & Methods
The study group was very diverse in terms of age, years of yoga practice, and comorbidities. This selection was intentional. Given this, were those who had practiced yoga for the shortest time already familiar with the breathing techniques used in this study?
Response: We acknowledge the pertinence of your question, which has helped identify a gap in our sample’s description. Thank you very much for your valuable comment. To recruit the sample for our study, a protocol was signed with the Portuguese Yoga Federation. The dissemination and recruitment of yoga practitioners were conducted by federation-certified instructors, who verified that the practitioners had already mastered the two yogic breathing techniques. Hence, the study only included yoga practitioners who had already achieved mastery of these techniques. The specified information has been appended and elucidated in the "Sample" subsection (lines 185-189).
Results
The results are clearly presented in seven tables and described in the text.
Response: Thank you for your valuable comment.
Discussion
The text in lines 416-419 should be removed. It would be helpful to start the discussion by reiterating the study's objective. Throughout the discussion, refer back to the objective and hypotheses (which should be added as mentioned above). Referring to tables in the discussion seems unnecessary. Alongside recommendations for future studies, it would also be useful to add a section on the limitations of the current study.
Response: Thank you for bringing the error to our attention. We have rectified the issue and removed the text referred to from our records.
After considering your feedback, we included a new paragraph at the beginning of the discussion to reinforce the study’s objectives and hypotheses, throughout the discussion (lines 451-457).
We appreciate your comments regarding the reference to the tables in the discussion. Nevertheless, we believe that given the number of variables, the table identification could assist the reader in redirecting their attention and focus, which could consequently facilitate the interpretation of the data. Consequently, we have decided to retain the identification of the tables.
As you suggested, we added a section on limitations following the recommendation for future studies. “For this study, it was not possible to access equipment that would allow moment-to-moment synchronized recording of respiratory rate, heart rate, interval between beats, blood pressure and peripheral oxygen saturation. This would enable a more precise analysis of the resonance between cardiac and respiratory systems, the specific influence of the relative duration of the respiratory cycle phases, and the deliberation of threshold values for a more significant acute positive impact on vital signs. Given the diversity of experience in practicing these respiratory techniques, we acknowledge that some more experienced practitioners automatically transitioned into meditation, which was not controlled” (lines 608-616).
Conclusions
I propose that in the conclusions section, the authors should not refer to other authors' studies (line 567) but rather focus solely on the conclusions drawn from their own research. The sentence in lines 574-576 is unnecessary in this section and should be placed in the “Recommendations for future studies” section.
Response: Thank you for your feedback. We have taken into consideration your input and have removed the reference to other authors' studies. The mentioned sentence has been relocated to the “Recommendations for future studies” section, as per your suggestion.
References
The bibliography includes as many as 90 references, which is excessive for research papers. It may be worthwhile to reduce the number of references, particularly those from before the year 2000.
Response: Thank you for your comment. We have made every effort to validate and enhance the entire manuscript with the references that we deemed most suitable for it. However, it is crucial to recognise that the total number of references may be excessive so, as recommended, we eliminated some older references, only keeping those classic and essential ones. For the same reason, we chose not to add additional references in the introduction. We also wish to inform you that the references included in this version were suggested by other reviewers, prompting our decision to incorporate them in comparison to the previous version.
Reviewer 2 Report
Comments and Suggestions for Authors
TITLE - Heart rate variability, blood pressure and peripheral oxygen saturation during yoga adham and mahat breathing techniques without retention in adult practitioners
General Comments
According to the title, the authors aim to characterize HRV, blood pressure and peripheral oxygen saturation basic yoga breathing slow techniques, with regular Yoga practitioners. The manuscript (MS) needs improvement and several clarifications before it can be suitable for publication. The MS is a bit hard to read; several sentences are poorly written. Extensive English editing is needed which I will not thoroughly address in my comments.
Specific Comments
Abstract
1. Lines 22-24 – This sentence is not clearly written.
2. Line 26 – Why did the authors only mention HRV in the aim when, in fact, they also used the other parameters mentioned in the title of the MS (BP and POS)?
3. Lines 38-39 – What do the authors mean with the expression “adults, mature adults and elderly”?
Introduction
4. Lines 53-55 – This sentence repeats what was previously said.
5. Lines 120-121 – “Breathing as a meditative practice in Yoga and an advanced practice called Pranayama,” – This part of the sentence is unintelligible.
6. Lines 151-153 – I believe this assumption is not entirely true. In a quick search I found these articles:
Heart Rate Variability Changes During and after the Practice of Bhramari Pranayama, L Nivethitha, NK Manjunath, and A Mooventhan
Investigating Components of Pranayama for Effects on Heart Rate Variability J Psychosom Res. 2021 Sep; 148: 110569, doi: 10.1016/j.jpsychores.2021.110569
Effects of Nadishodhana and Bhramari Pranayama on heart rate variability, auditory reaction time, and blood pressure: A randomized clinical trial in hypertensive patients, Journal of Ayurveda and Integrative Medicine Volume 14, Issue 4, July–August 2023, 100774
Immediate effect of Kapalbhathi pranayama on short term heart rate variability (HRV) in healthy volunteers.S. Lalitha , K. Maheshkumar , R. Shobana EMAIL logo and C. Deepika Journal of Complementary and Integrative Medicine https://doi.org/10.1515/jcim-2019-0331
7. Lines 155-158 – This sentence is not clear.
Methods
8. Line 167 – What do the authors mean with “due to limited condition”?
9. Line 169 - Why did the authors considered only one person to take the withdrawal into account? That is a very small withdrawal rate 2.2%)! What was this withdrawal rate calculated on?
10. Line 172 and others – It is better to replace the word “gender” with the word “sex”.
11. Lines 199-204 – This information would fit better in the Results section.
12. Lines 236-239 - This sentence does not sound right to me.
Results
13. Line 266 - What do the authors mean with “abdominal conditions”? Wouldn't it be better to say “abdominal breathing”?
14. Line 278 - Remove “This is a table. Tables should be placed in the main text near to the first time they are cited” and put the title of Table 3.
15. Line 280 - The asterisk should follow the word “conditions”
16. Lines 304-307 - This sentence is not well constructed.
17. Line 336 – Replace “complete technique” with “complete breathing”.
18. Line 368 – Replace “Table 76” with “Table 6”.
Discussion
19. Lines 416-419 - Delete the instruction “Authors should discuss the results and how they can be interpreted from the perspective of previous studies and of the working objectives. The findings and their implications should be discussed in the broadest context possible. Future research directions may also be highlighted.”
20. Lines 421-422 – Please, explain. This is not clear in the Results, by looking to Table3, as the authors suggest.
21. Lines 432-435 - This sentence is confusing.
22. Lines 442-444 - This not clearly written.
23. Lines 444-446 - This is not mentioned in Table 3 as the authors state. What happen to the values if you take out these outliers? Does this mean that hypertensive patients should not practice these pranayamas?
24. Lines 446 – “Perhaps white coat effect was present” - Why? Did it only happen to these women?
25. Lines 449-451 – This sentence is not clear.
26. Line 454 – “Breathing frequency has effects on lungs gas exchange, with higher efficiency around 6 breaths per minute (0.1 Hz) [75].” Where is this mentioned in the reference cited (75)? I did not find it. And, if 6 breaths per minute have a higher efficiency on lungs gas exchange, why is bradypnea considered a symptom of an underlying health condition? This issue needs to be discussed.
27. Lines 458-459 – “with nearer values that are expected in a healthy adult person” – What do the authors mean?
28. Line 463 – “many years of yoga practice” - Why didn't the authors use two groups, one with fewer years of practice and the other with more years of practice? There is a wide variation in the number of years of practice, which makes it difficult to interpret the results.
29. Lines 464-465 - And what about the confounding effect of more years of practice?
30. Lines 466-469 - What do the authors conclude from these data?
31. Lines 505-508 - I would like to see a comment on this published statement: "We do not recommend the use of the LF/HF metric to measure autonomic balance during slow breathing. The LF/HF metric is most useful during sleep when the respiratory rate is typically high enough for vagal activity to manifest in the HF band of the HRV power spectrum". (Heart rate variability during mindful breathing meditation Aravind Natarajan, Front Physiol. 2022; 13: 1017350. doi: 10.3389/fphys.2022.1017350)
Conclusions
32. Lines 569-572 – “The complete technique revealed to produce superior effects; although, due to it greater motor control complexity, abdominal technique is recommended initially, as it yields similar physiological benefits in the analyzed vital signs.” – Poorly constructed sentence; it is confusing which technique involves greater control complexity.
Comments on the Quality of English Language
I have already mention this in my review.
Author Response
Dear reviewer,
We are most grateful for your constructive and thought-provoking suggestions, which are contributing to the enhancement of the quality of the information in the manuscript.
We have carefully considered all your comments and made the appropriate changes. Below, you will find our responses to each of your comments, presented in the order they were received. To facilitate the review process, we have highlighted our responses and the corresponding changes in the manuscript in green.
General Comments
According to the title, the authors aim to characterize HRV, blood pressure and peripheral oxygen saturation basic yoga breathing slow techniques, with regular Yoga practitioners. The manuscript (MS) needs improvement and several clarifications before it can be suitable for publication. The MS is a bit hard to read; several sentences are poorly written. Extensive English editing is needed which I will not thoroughly address in my comments.
Response: As suggested, the manuscript was proofread in its entirety by a native English speaker, bachelor in Languages and Humanities, in order to make it easier to read and understand. These linguistic changes are highlighted in dark orange in the manuscript.
Specific Comments
Abstract
- Lines 22-24 – This sentence is not clearly written.
Response: Thank you for your comment. We have revised and clarified the sentence.
- Line 26 – Why did the authors only mention HRV in the aim when, in fact, they also used the other parameters mentioned in the title of the MS (BP and POS)?
Response: Thank you for your question, in fact we wanted to characterise several vital signs, that were inadvertently omitted. We have incorportated these into the sentence in question (line 26).
- Lines 38-39 – What do the authors mean with the expression “adults, mature adults and elderly”?
Response: The use of the terms 'adults,' 'mature adults,' and 'elderly' in the abstract, was employed with the intention of emphasising the age-related comprehensive nature of our findings. The sample was not limited to 'adults'; it included practitioners aged between 32 and 78 years. This range encompasses 'adults' aged 18 to 50, 'mature adults' aged 50 to 65 and 'elderly' individuals aged 65 and over. We believe that this terminology enhances the value of the results obtained.
Introduction
- Lines 53-55 – This sentence repeats what was previously said.
Response: Thank you for your comment, we have removed the sentence.
- Lines 120-121 – “Breathing as a meditative practice in Yoga and an advanced practice called Pranayama,” – This part of the sentence is unintelligible.
Response: In light of the aforementioned comments, the sentence in question has been revised and clarified in the manuscript: “Breathing as a meditative practice in Yoga and an advanced practice called Pranayama, through breath control and expansion with retention, allows conscious regulation of breathing rate, depth and/or the inhalation/exhalation ratio” (lines 128-130).
- Lines 151-153 – I believe this assumption is not entirely true. In a quick search I found these articles:
Heart Rate Variability Changes During and after the Practice of Bhramari Pranayama, L Nivethitha, NK Manjunath, and A Mooventhan Investigating Components of Pranayama for Effects on Heart Rate Variability J Psychosom Res. 2021 Sep; 148: 110569, doi: 10.1016/j.jpsychores.2021.110569
Effects of Nadishodhana and Bhramari Pranayama on heart rate variability, auditory reaction time, and blood pressure: A randomized clinical trial in hypertensive patients, Journal of Ayurveda and Integrative Medicine Volume 14, Issue 4, July–August 2023, 100774
Immediate effect of Kapalbhathi pranayama on short term heart rate variability (HRV) in healthy volunteers.S. Lalitha , K. Maheshkumar , R. Shobana EMAIL logo and C. Deepika Journal of Complementary and Integrative Medicine https://doi.org/10.1515/jcim-2019-0331
Response: Thank you for your comment and the articles you suggested. As we also mentioned to reviewer 1, and in our introduction, one of the main purposes of our study was to analyse the effect of breathing modes per se, without including vertical postures, specific breathing techniques (such as alternately closing the nostrils or humming), imposition of breathing frequency, or vigorous exhalation. We are aware that there have been previous studies investigating the effect of yoga breathing techniques on vital signs. However, to the best of our knowledge, and including these identified studies, no previous analysis has considered such a comprehensive range of vital signs, nor peripheral oxygen saturation. The various studies attribute the improvements observed to an enhanced efficiency of the cardiorespiratory system, suggesting in some cases a more efficient functioning of gas exchange in the lungs. However, they did not assess oxygen saturation. By analysing the different variables (i.e., heart rate, heart rate variability, respiratory rate, blood pressure and peripheral oxygen saturation), our study provides a specific characterisation of the two techniques: (adham pranayama and mahat yoga pranayama), which are probably universally available. This contributes to a deeper understanding of their acute effects on vital signs. We have attempted to clarify this information in the identified paragraph. which reinforces the originality and strength of our study (lines 161-174).
- Lines 155-158 – This sentence is not clear.
Response: Thank you for your comment, we have changed the sentence to make it clearer.
Methods
- Line 167 – What do the authors mean with “due to limited condition”?
Response: Thank you for your comment, we have clarified the sentence. The phrase used referred to the difficulty in articulating schedules between experimenters, yoga teachers and participants. We have removed the sentence.
- Line 169 - Why did the authors considered only one person to take the withdrawal into account? That is a very small withdrawal rate 2.2%)! What was this withdrawal rate calculated on?
Response: We are grateful for your inquiry. It became evident that the wording of our original statement was not sufficiently clear. During the sampling process, we attempted to engage a larger number of participants than was initially anticipated, based on the sample calculation. However, not all of the yoga practitioners who provided informed consent were available during the data collection period. Furthermore, five of the participants who were mobilized ultimately withdrew from the study, two due to individual constraints in one or two of the breathing conditions and one due to illness. This information has been clarified in the text (lines 181-184).
- Line 172 and others – It is better to replace the word “gender” with the word “sex”.
Response: The word “gender” has been replaced by the word “sex” throughout the entire manuscript.
- Lines 199-204 – This information would fit better in the Results section.
Response: We have followed the suggestion and moved the identified text back to the results section.
- Lines 236-239 - This sentence does not sound right to me.
Response: We are grateful for your feedback and have revised the sentence in question to enhance clarity and comprehension.
Results
- Line 266 - What do the authors mean with “abdominal conditions”? Wouldn't it be better to say “abdominal breathing”?
Response: We are grateful for your proposal and have implemented abdominal breathing as an alternative.
- Line 278 - Remove “This is a table. Tables should be placed in the main text near to the first time they are cited” and put the title of Table 3.
Response: We are grateful for your observation. The error has been rectified, and the correct title has been entered.
- Line 280 - The asterisk should follow the word “conditions”
Response: All tables have been revised.
- Lines 304-307 - This sentence is not well constructed.
Response: We would like to thank you for bringing this to our attention. We have rewritten the sentence in question.
- Line 336 – Replace “complete technique” with “complete breathing”.
Response: Change made.
- Line 368 – Replace “Table 76” with “Table 6”.
Response: We would like to thank you for bringing this to our attention. The table number in question has indeed been changed.
Discussion
- Lines 416-419 - Delete the instruction “Authors should discuss the results and how they can be interpreted from the perspective of previous studies and of the working objectives. The findings and their implications should be discussed in the broadest context possible. Future research directions may also be highlighted.”
Response: This sentence has been inadvertently omitted; it has since been removed.
- Lines 421-422 – Please, explain. This is not clear in the Results, by looking to Table3, as the authors suggest.
Response: Please be advised that the information in question pertains to Table 2. The requisite correction has been made.
- Lines 432-435 - This sentence is confusing.
Response: We agree with your comment, the sentence was somewhat opaque. We have revised it to enhance its intelligibility.
- Lines 442-444 - This not clearly written.
Response: The sentence has been rewritten in order to provide greater clarity.
- Lines 444-446 - This is not mentioned in Table 3 as the authors state. What happen to the values if you take out these outliers? Does this mean that hypertensive patients should not practice these pranayamas?
Response: Your comment and queries are duly noted and appreciated. Please be advised that the data referenced in your question is not included in Table 3. Consequently, we have removed it from the sentence. As a consequence of the principle of universality assumed in the composition of the sample, we detected this occurrence. In our studies, the approach is to include outliers, as the assumption is that if the population is diverse, it is important to consider all individuals within that population. As a consequence of these two cases, it is evident that the sample is insufficient to generalise the findings to a population with similar characteristics (e.g., isolated systolic hypertension or total hypertension, without medication). Therefore, we consider it convenient to monitor blood pressure during the practice of these techniques. At present, we are not in a position to say definitively that hypertensive patients without medication should not practice these pranayamas, namely because we had medicated hypertensive patients in the sample who did not reveal this pattern. The hypertensive patients mentioned were not aware of their condition, nor were they taking medication for this purpose. Therefore, we consider it important to monitor blood pressure while practicing these techniques, as a preventive measure. Nevertheless, these results justify the pertinence of new studies with these yogic techniques in hypertensive participants. It is for this reason that the findings are being shared with the scientific community.
Moreover, we wish to announce that we are engaged in data processing with a select group of medicated individuals within the sample. For those diagnosed with hypertension, we can provide additional information that was not included in the article.
A comparative analysis of participants without other health complications and non-smokers (T) (N= 19, ♀=17, 50.47±11.38 years of age, 11.37±10.66 years of practice), with participants medicated for hypertension (H) (N=8, ♀=6, 1 smoker, 68.25±7.36 years of age, 8.91±9.34 years of practice). Results: In the baseline (B), H with higher systolic (PS)* and diastolic (PD)* pressure, lower SpO*, SD2*, SDNN*, pNN50* and HRV index*; in abdominal breathing (A), H with higher PD*; and, in complete breathing (D) with lower SD2*, than T. Compared to B, during breathing techniques, H reduced respiratory frequency (RF)*** (to 3-4cpm) and PS by 6-7 mmHg (ns), increased SpO*, SD2*, HRV index (A*, D**), and HFn (A *, D**); during A, H increased standard deviation of heart rate*; and, during D, H increased LF/HF* ratio. Conclusions: The notable discrepancies observed between H and T in B, are largely diminished during the execution of A and D, yielding pronounced positive outcomes in terms of SpO, PS and HRV (in comparison to the baseline).This is despite the considerable age disparity between H (the older group) and T (the younger group). The reduction in PS may indicate the activation of baroreflex sensitivity. The application of RF at 0.07-0.05 Hz resulted in an increase in VLF (ns), which could be indicative of resonance.
- Lines 446 – “Perhaps white coat effect was present” - Why? Did it only happen to these women?
Response: To our knowledge, only these two women have been affected. The detection was made during data processing, as consistent outliers in the box plots of the different breathing conditions. As neither of them has a medical diagnosis of hypertension (we considered them as probably hypertensive due to their baseline values), we considered the hypothesis of a white coat effect. To validate this hypothesis, it would be necessary to conduct additional longitudinal blood pressure measurements, a procedure that was not foreseen in the experimental design. This information has been incorporated into the manuscript.
- Lines 449-451 – This sentence is not clear.
Response: We would like to thank you for your comment. We recognise that the original sentence was unclear and that a comparison of systolic arterial pressure was not made. Consequently, we have removed the sentence to avoid any further confusion.
- Line 454 – “Breathing frequency has effects on lungs gas exchange, with higher efficiency around 6 breaths per minute (0.1 Hz) [75].” Where is this mentioned in the reference cited (75)? I did not find it. And, if 6 breaths per minute have a higher efficiency on lungs gas exchange, why is bradypnea considered a symptom of an underlying health condition? This issue needs to be discussed.
Response:
We offer our apologies for the unintentional inclusion of an inaccurate reference. The required information can be found in the article referenced below, which we have duly updated in the manuscript.
Lehrer, P. M., Vaschillo, E., Trost, Z., & France, C. R. (2009). Effects of rhythmical muscle tension at 0.1 Hz on cardiovascular resonance and the baroreflex. Biological Psychology, 81, 24–30. doi: 10.1016/j.biopsycho.2009.01.003
The occurrence of bradypnea in an involuntary manner may frequently indicate the existence of an underlying health issue. In contrast to slow breathing exercises, which are intentionally motor - controlled, as exemplified by yoga practice, non-intentional bradypnea can potencially result in inadequate oxygenation and an elevated carbon dioxide concentration, indicating that the body’s innate ability to regulate respiration may be diminished. Nevertheless, it is of great importance to analyse bradypnea within the context of the individual, whether it is intentional or non-intentional, as well as to ascertain whether it represents good or bad oxygenation. Another example that illustrates the necessity for analysis of the individual’s context is that of athletes. It can be expected that an ultramarathon runner will exhibit a reduction in breathing frequency during periods of rest. In this specific case, the bradypnea can be attributed to the enhanced efficiency of the respiratory and cardiovascular systems, rather than to a pathological condition.
- Lines 458-459 – “with nearer values that are expected in a healthy adult person” – What do the authors mean?
Response: The sample was composed of adults with a wide age range, including the elderly (who typically exhibit lower oxygen saturation levels), as well as individuals with and without pathological conditions (some pathologies also result in reduced oxygenation). In the light of the aforementioned, it is eident that the breathing techniques associated with yoga are capable of overcoming the initial disadvantages pertaining to peripheral oxygen saturation, with the observed values approaching those anticipated for a typical healthy adult population
Please note response 23, which includes data treatment for comparison between the medicated hypertensive and healthy participants in this sample. This additionally reinforces our initial statement.
- Line 463 – “many years of yoga practice” - Why didn't the authors use two groups, one with fewer years of practice and the other with more years of practice? There is a wide variation in the number of years of practice, which makes it difficult to interpret the results.
Response: The composition of the sample was designed in line with the principle of universality, whereby the intention was to create a diverse sample with regard to a number of predefined variables (e.g., gender, experience, presence of health problems, age). The primary objective was to assess the impact of yogic breathing techniques. If this impact is found to be significant in a diverse population, which is also more representative of the typical yoga practitioner, this would serve to reinforce the efficacy of the techniques. In other words, it was not necessary to isolate physiological variables in small groups to observe differences. The technique produces a significant effect that is evident even in such a diverse sample. Consequently, and because it was not the objective of the study, we elected not to categorise the practitioners according to their level of experience.
- Lines 464-465 - And what about the confounding effect of more years of practice?
It is recommended that the supplementary data treatment outlined in question 23 be taken into consideration. It appears that the potential confounding effect of “Age”, is diminished during the implementation of these breathing techniques, as evidenced by the case of medicated hypertensives, who constituted a significantly older age group than the healthier participants. Nevertheless, the breathing techniques yielded comparable SpO2 and a reduction in systolic pressure.
- Lines 466-469 - What do the authors conclude from these data?
A commentary has been included in the conclusions section, which states: “It would be beneficial for yoga instructors to monitor the individual impact of blood pressure during the practice of these techniques, in order to ascertain whether the desired effect is not occurring for particular individuals. It may also be advantageous to explore individualized resonance frequency rates inspecific cases.”
- Lines 505-508 - I would like to see a comment on this published statement: "We do not recommend the use of the LF/HF metric to measure autonomic balance during slow breathing. The LF/HF metric is most useful during sleep when the respiratory rate is typically high enough for vagal activity to manifest in the HF band of the HRV power spectrum". (Heart rate variability during mindful breathing meditation Aravind Natarajan, Front Physiol. 2022; 13: 1017350. doi: 10.3389/fphys.2022.1017350)
Response: We concur with the author’s viewpoint in its entirety. Our data corroborate his reflections, which we share primarily in lines 488-495 and also in lines 518-531. It is of paramount importance to note that during slow, controlled breathing, predominant frequencies are LF and VLF. This indicates that the ratio LF/HF tends to have a higher value, significantly higher than 1. Therefore, for the analysis of the effect of these kinds of voluntarily controlled breathing techniques, it would be beneficial to consider additional parameters, particularly those representing the process (and not exclusively the product) of vital signs changing. These could include qualitative and quantitative analysis of Poincaré plots, or other non-linear analysis, such as recurrence, which we intend to explore in future research.
We are grateful for your recommendation and had selected to incorporate it into the text at the indicated location.
Conclusions
- Lines 569-572 – “The complete technique revealed to produce superior effects; although, due to it greater motor control complexity, abdominal technique is recommended initially, as it yields similar physiological benefits in the analyzed vital signs.” – Poorly constructed sentence; it is confusing which technique involves greater control complexity.
Response: Thank you, we have rephrased the sentence to make it clearer: “The full technique was found to yield superior effects. However, due to its greater motor control complexity, the abdominal technique is recommended initially, as it produces similar physiological benefits in the vital signs analysed”.
Reviewer 3 Report
Comments and Suggestions for Authors
Dear authors,
The manuscript is interesting, but some issues must be addressed prior to acceptance.
1. The introduction is way too large. Please, attain to the strictly necessary to provide readability.
2. Why did you set the correlation in 0.7 for sample size calculation? Clarufy.
3. Table 1 should be summarized in frequencies (%) to improve readability.
4. The study is a quasi-experimental design, right?
5. "which is validated for adults" - Please, insert the device's psychometric values.
6. Any post hoc analysis for ANOVA? Explain how did you correct the data to avoid multiple testing.
7. Insert this statement in the results section: Due to technical problems, data of 256 peripheral oxygen saturation of two participants were lost.
8. Consider less tables to your results section. Although a researcher would be able to follow them, a clinician would experience some issues for that. I strongly recommend you to attain to the differences, and include other data in a supplementary material. Be concise.
9. I could not find a limitations section. Please, include and clarify.
10. Make a statement for data availability. I strongly advise you to upload the raw data in an open online repository, such as Mendeley Data.
Author Response
Dear reviewer,
Thank you very much for taking the time to review this manuscript. Your valuable feedback has significantly enhanced and enriched the content.
We have given your comments thorough consideration and have made the required revisions accordingly. Below, you will find our responses to each of your comments, presented one by one. To facilitate the review process, we have highlighted our responses and the corresponding changes in the manuscript in purple.
Dear authors,
The manuscript is interesting, but some issues must be addressed prior to acceptance.
- The introduction is way too large. Please, attain to the strictly necessary to provide readability.
Response: Thank you for your comment. However, we are confident that the main issue with the introduction’s readability, was not its length but rather its lack of fluency due to the English language. In fact, another reviewer suggested that we expand the introduction by reviewing additional studies. We offer our sincere apologies, but we have elected to retain the introduction, which encompasses pivotal studies pertinent to our main objective and methodology.
Furthermore, in accordance with the recommendations of the peer reviewers, the entire manuscript was subjected to a thorough review by a native English speaker, to guarantee the fluency and readability of all sections. The language changes are highlighted in the text in dark orange.
- Why did you set the correlation in 0.7 for sample size calculation? Clarufy.
Response: We are grateful for your inquiry. The correlation between the measures was set at 0.7 based on a previous study, that analysed heart rate variability in yoga breathing techniques. This rationale has been explicitly stated and clarified in the manuscript (lines 180-181).
- Table 1 should be summarized in frequencies (%) to improve readability.
Response: It is our considered opinion that the presentation of the frequency of health issues among our participants will enhance the readability of the data. In accordance with both this proposal and your suggestion in comment number 7, in the relevant information has been incorporated into the text, and also included in Supplementary Table A1. This approach allows readers to readily identify the distinctive attributes of the individuals depicted in the text, while also facilitating access to a more detailed characterization in the supplementary table (lines 658-663).
- The study is a quasi-experimental design, right?
We are grateful for your observation. We may consider it as a quasi-experimental study, as we have information about cause and effect, through an intervention (though not a treatment, in the classical sense), which allows us to formulate objective hypotheses. We made the necessary correction (lines 207-209).
- "which is validated for adults" - Please, insert the device's psychometric values.
Response: We are grateful for your valuable input and have incorporated the psychometric values for the Polar V800. A previous validation study demonstrated that the device has a combined error rate of 0.086% and an Intraclass Correlation Coefficient (ICC) exceeding 0.999 (lines 229-230).
- Any post hoc analysis for ANOVA? Explain how did you correct the data to avoid multiple testing.
Response: We are grateful for your observation. In fact, no ANOVA is presented, we neglected to remove this sentence. It has now been removed.
- Insert this statement in the results section: Due to technical problems, data of 256 peripheral oxygen saturation of two participants were lost.
Response: As recommended, the sentence in question has been replaced in the results section.
- Consider less tables to your results section. Although a researcher would be able to follow them, a clinician would experience some issues for that. I strongly recommend you attain to the differences, and include other data in a supplementary material. Be concise.
Response: We are grateful for your suggestion. In light of this, we have opted to relocate tables 1 and 4 to appendices, thus streamlining the manuscript and providing readers with convenient access to the supplementary tables if they so desire (lines 658-666).
- I could not find a limitations section. Please, include and clarify.
Response: We are grateful for your valuable input and have incorporated it into the revised version of the study, which now includes a limitations section that provides a detailed account of the study’s constraints (lines 607-616).
- Make a statement for data availability. I strongly advise you to upload the raw data in an open online repository, such as Mendeley Data.
Response We appreciate the recommendation. However, as the author team is still engaged in additional data analysis for future submissions, we have chosen to refrain from making the data public at this time. Furthermore, due to ethical and legal restrictions imposed by our ethics committee and institutional guidelines, data sharing is only possible upon request and with the superior approval. Nevertheless, in the meantime, we have provided all the information necessary for an article to be selected for systematic review and a meta-analysis, in line with the expectations of the journal.
A statement has been included in the text which states that data availability is possible upon request and with the approval of the relevant authorities (lines 653-654).
Round 2
Reviewer 3 Report
Comments and Suggestions for Authors
Dear authors,
Thank you for all changes to enhance the readability.
I have a single issue to address: decimals use "." instead of ",". Please, be aware or that throughtout the entire manuscript.
Author Response
Dear reviewer,
Thank you very much for all your comments. As you suggested, we have changed the decimal separators that were mistakenly “,” to “.”.
Best Regards